# Monogamy relations for relativistically causal correlations

**Mirjam Weilenmann**  ✉

Non-signalling conditions encode minimal requirements that any (quantum) systems must satisfy in order to be consistent with special relativity. Recent works have argued that in scenarios involving more than two parties, correlations compatible with relativistic causality do not have to satisfy all possible non-signalling conditions but only a subset of them. Here we show that correlations satisfying only this subset of constraints have to satisfy highly non-local monogamy relations between the effects of space-like separated random variables. These monogamy relations take the form of entropic inequalities between the various systems and we give a general method to derive them. Using these monogamy relations, we refute previous suggestions for physical mechanisms that could lead to relativistically causal correlations, demonstrating that such mechanisms would lead to superluminal signalling.

The relation between quantum and relativity theory has been a topic of interest in the realm of quantum information theory in the last few decades. Investigations into whether consistency with special relativity is sufficient to restrict correlations between two parties to be quantum[1], has prompted extensive research in quantum foundations into the physical principles underlying it. At the same time, cryptographic tasks have been analysed against adversaries that are not necessarily restricted to act according to quantum theory but instead are restricted by relativistic principles captured by non-signalling constraints; examples of such tasks include device-independent key distribution[2] and randomness extraction[3,4].

As quantum technologies progress to involve larger systems and more parties, e.g., in quantum networks, the question of generalising these concepts becomes increasingly important. It was argued early on that the straightforward generalisation of non-signalling to more parties[5] can be relaxed in specific setups. That is, if such a relaxation does not introduce causal inconsistencies such as causal loops[6,7]. This led to a general notion of relativistically causal correlations[8].

Subsequent work showed that information-theoretic and statistical challenges may arise when restricting correlations in the multiparty setting only by this notion of relativistic causality. Specifically, it leads to no-go theorems for cryptographic protocols[9] as well as to challenges for causal modelling, which in this case relies on fine-tuning causal dependencies[10,11].

Furthermore, correlations alone do not explain, what mechanisms they might arise from. For understanding in what sense correlations can be generated in an information theoretic task or used by an adversary to tamper with a cryptographic protocol, these mechanisms and the control different parties may have over them are important. Despite the lack of a full explanation, a partial intuition of such a mechanism was given in ref. 7, where there is an agent, the jammer, "who has access to a jamming device which he can activate, at will"[7].

In this work, we demonstrate conceptual implications of allowing the most general relativistically causal correlations. While the examples considered in refs. 6–9 seem to suggest that causality is relaxed in a somewhat innocent way, we show that in more complicated multiparty setups, these correlations have to satisfy a type of monogamy relation. The special feature of these relations is that they relate the influences that independent, space-like separated variables can have. We further show, that our monogamy relations can be used to rule out the explanation of correlations in terms of a jamming mechanism, as such control would enable a superluminal transfer of information. We discuss further implications of the monogamy relations, including potential other explanations of relativistically causal correlations and implications for causal modelling. Our monogamy relations can be phrased in terms of entropic inequalities for which we propose a technique to derive them and compare entropic constraints on relativistically causal correlations to their non-signalling analogues[12].

## Results

The present work is concerned with the constraints imposed by relativistic causality on the correlations observed by multiple parties[6–8].

Département de Physique Appliquée, Université de Genève, Genève, Switzerland. ✉e-mail: mirjam.weilenmann@unige.ch

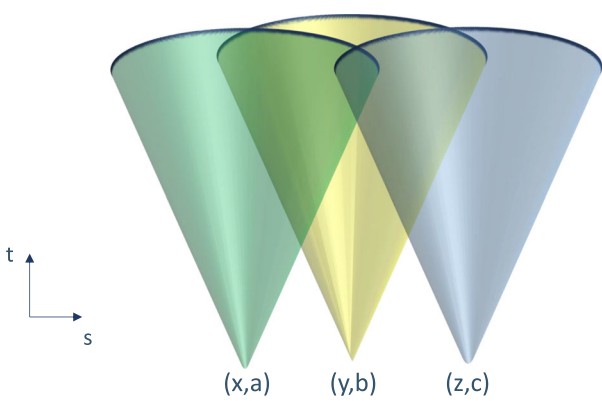

**Fig. 1 | Contrasting multipartite generalisations of non-signalling conditions.** Three-party scenario involving Alice, Bob and Charlie with inputs (outputs) $x$, ($a$), $y$, ($b$) and $z$, ($c$), respectively. The parties are arranged on a line such that their choice of input and generation of output happens space-like separated, as illustrated with the three future light cones. As the intersection of the light cones $\mathcal{K}_{(x,a)}$ and $\mathcal{K}_{(z,c)}$ lies fully within $\mathcal{K}_{(y,b)}$, the relativistic causality constraints do not include $P(ac|xyz) = P(ac|xz) \; \forall \, a, c, x, y, z$, which is among the usual non-signalling constraints[5]. Notice that this means that Bob's input can only affect the correlations of Alice and Charlie not their marginals, i.e., $P(a|xyz) = P(a|x)$, $P(c|xyz) = P(c|z) \; \forall \, a, c, x, y, z$. One way to achieve such an influence is by means of the functional dependency $A \oplus C = Y$, which will be used throughout this work.

We mean by relativistic causality the requirement that no observer can send superluminal signals, in the sense that an input $X$ can influence correlations of a set of variables $\{A_i\}_i$, if and only if the intersection $\cap_i \mathcal{K}_{a_i}$ of the future light cones $\mathcal{K}_{a_i}$ of these variables lies fully in the future light cone of $X$, otherwise it cannot, i.e., otherwise $P(\{a_i\}_i|x) = P(\{a_i\}_i) \forall \{a_i\}_i, x$. This arguably corresponds to the notion captured by the mathematical formalisation from refs. 7–9. The standard tripartite example comparing these constraints to the full non-signalling constraints was previously considered in refs. 7,8, and is presented in Fig. 1. The notion of relativistic causality captures the intuition that having variables affect correlations of other variables does not lead to logical contradictions whenever these correlations can only be evaluated in the future of the former variables[7,8]. In the example of Fig. 1, this is the case for the variables $A$ and $C$, the correlations of which can only be seen in the future of $Y$.

Formally, we define the constraints imposed by relativistic causality as follows.

**Definition 1 (Relativistic causality constraints).** Let there be $n$-parties labelled $1, \ldots, n$ and let there be a strict subset of the parties $\mathcal{J} \subset \{1, \ldots, n\}$ and let $\mathcal{J}^c = \{1, \ldots, n\} \setminus \mathcal{J}$. Let $\mathcal{A}_\mathcal{J} = \{a_i | i \in \mathcal{J}\}$ and $\mathcal{X}_\mathcal{J} = \{x_i | i \in \mathcal{J}\}$ and define $\mathcal{X}_\mathcal{K}$ to be the set of $x_k \in \mathcal{X}_{\mathcal{J}^c}$ such that the intersection of the future of all the variables in $\mathcal{A}_\mathcal{J}$, $\cap_{j \in \mathcal{J}} \mathcal{K}_{a_j}$, lies within the future of $X_k$. Then, $\forall \, \mathcal{J}$,

$$\sum_{a_i : i \in \mathcal{J}^c} P(a_1, \ldots, a_n | x_1, \ldots, x_n) = P(\mathcal{A}_\mathcal{J} | \mathcal{X}_\mathcal{J} \mathcal{X}_\mathcal{K}), \forall a_1, \ldots, a_n, x_1, \ldots, x_n \quad (1)$$

Notice that scenarios where only a subset of the involved parties are considered are obtained by marginalising over the other parties (outcomes). In the following, we call correlations that satisfy all relativistic causality constraints in a light-cone arrangement the relativistically causal correlations of that scenario.

While this formal definition is stated straightforwardly, the physical notion behind it deserves a few additional remarks. First, relativistically causal correlations differ from general multipartite non-signalling correlations[5] (see Fig. 1 for an example).

Second, this notion is incompatible with the operational reasoning that is usually followed in quantum information theory (or in

treatments in quantum foundations, like generalised probabilistic theories[13,14]), where the non-signalling conditions are naturally built into the operational framework. In the quantum case this is done by taking space-like separated measurements to be made up from commuting operators or to act on different Hilbert spaces in a tensor product. However, it was suggested that relativistically causal correlations might appear in future physical theories[8].

Thirdly, the relativistic causality constraints are frame-independent, because any event in the future light cone of some event is in this future light cone in any reference frame. The analogous statement holds for events in the intersection of several light cones. We discuss this as well as the dependency of the relativistic causality constraints on the number of spatial dimensions of the setup in Supplementary Note 1.

While the set of relativistically causal correlations is thus straightforwardly defined, an underlying explanation, i.e., a mechanism leading to such correlations is not known and left open in previous work[7,8]. The intuition that is given is that an agent can activate a device that jams correlations of other parties[7]. While in ref. 7 this mechanism is instantaneous, it is considered in terms of superluminal influences that travel at finite speeds and considered among a larger number of parties in ref. 8.

Note that while the relativistic causality constraints remain valid independently of the frame, these underlying explanations depend on the reference frame, as the condition that a signal from a spacetime point $Z$ travelling at a speed $u > c$ can reach a point $A$ is not Lorentz invariant. Indeed, if a signal can travel this way in one reference frame, there may be another reference frame where the order of events at $Z$ and $A$ is reversed. Thus, such an explanation can only be retained in certain reference frames[7] and may be different in others.

Notice that these explanations specifically aim to explain the mechanism that leads to superluminal influences beyond standard non-signalling. For this, an explanation in the sense of the quantum formalism for quantum correlations (or a generalised probabilistic theory for non-signalling correlations) is not strong enough (as both abide by non-signalling). Thus, an explanation or mechanism for these correlations has to be of a more general nature; this might involve some effect that propagates this influence (in the spirit of the jamming intuition above) or might have to be of an as yet inconceivable nature.

Finding such a mechanism is thus a general problem, that differs from attempting to replace the usual quantum (or generalised probabilistic) explanation of quantum (or non-signalling) correlations by means of hidden communication[15–19]. One could, however, attempt to use features in the spirit of a hidden communication mechanism in combination with existing theories to explain relativistically causal correlations. However, the set of relativistically causal correlations cannot arise solely from hidden communication models where influences propagate at finite speeds[8,15–19].

In the following subsections, we present our results for relativistically causal correlations.

## Monogamy of relativistically causal correlations

In this section, we show that for scenarios with more than three parties, the relativistically causal correlations have to satisfy strong monogamy constraints. We show this by means of the setup in Fig. 2 and refer to the Supplementary Information for other examples and to the "Methods" for a general method to formulate entropic monogamy inequalities.

Thus, let us consider the setting illustrated in Fig. 2. This scenario allows us to isolate the effect of non-local relativistically causal influences from local ones by denoting only either input or outcome at each space-time point and ignoring the other. This means that correlations are solely established via non-local effects. In this scenario, the

(a)

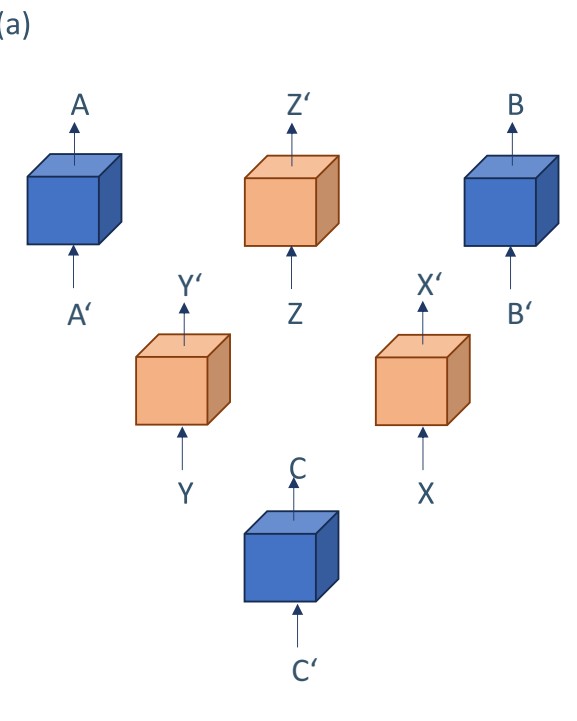

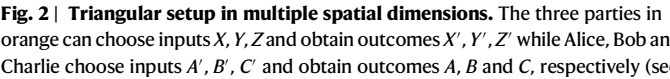

(b)

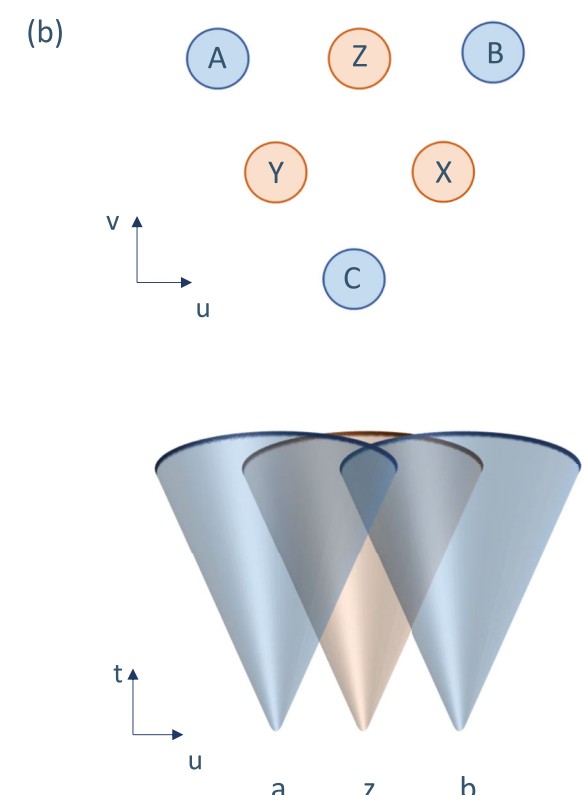

**Fig. 2 | Triangular setup in multiple spatial dimensions.** The three parties in orange can choose inputs $X, Y, Z$ and obtain outcomes $X', Y', Z'$ while Alice, Bob and Charlie choose inputs $A', B', C'$ and obtain outcomes $A, B$ and $C$, respectively (see (**a**)). For the correlations considered in the main text, only $A, B, C, X, Y, Z$ play a role. The six events where these variables are generated are space-like separated and are displayed here in a frame where they are simultaneous (see (**b**)).

light-cone arrangement imposes, according to Definition 1, the relations

$$P(ab|xyz) = P(ab|z)$$
$$P(ac|xyz) = P(ac|y)$$
$$P(bc|xyz) = P(bc|x)$$
$$P(a|xyz) = P(a)$$
$$P(b|xyz) = P(b)$$
$$P(c|xyz) = P(c),$$

(2)

$\forall\ a, b, c, x, y, z$, which give rise to the set of relativistically causal correlations

$$\mathcal{K} = \{P | P(abc|xyz) \geq 0\ \forall a, b, c, x, y, z,$$
$$\sum_{a,b,c} P(abc|xyz) = 1\ \forall x, y, z$$
$$\text{and the equalities (2) hold}\}.$$

(3)

Performing a vertex enumeration, we can express any correlations $P \in \mathcal{K}$ as a convex combination of the finite number of extremal vertices of the polytope $\mathcal{K}$. For the case of binary $a, b, c, x, y, z$, we have performed this computation with the software PORTA[20]. This leads to a set of correlations that is spanned by 11964 extremal vertices. Out of these, we find that of the three constraints

$$P(ab|xyz) = P(ab),$$ (4)

$$P(ac|xyz) = P(ac),$$ (5)

$$P(bc|xyz) = P(bc),$$ (6)

$\forall\ a, b, c, x, y, z$ which have been dropped compared to the standard non-signalling constraints[5] (see also Supplementary Equation (S1)), 8624 extremal vertices violate all three, 3·968 violate two of them, 3·60 violate one and 256 violate none. Due to the large number of extremal vertices, we do not list them here.

An example of a compatible distribution is constructed using the following functional dependency of variables: In this setup, it is possible that $Z$ determines whether $A$ and $B$ are correlated or anti-correlated, i.e., $Z = A \oplus B$. This in turn implies that $X$ and $Y$ cannot be causally connected to the correlations of $B$, $C$ or $A$, $C$ respectively. Indeed, combining the constraints specifying $\mathcal{K}$ and $Z = A \oplus B$ for the marginal $P(ab|z)$, we find that all compatible distributions satisfy $P(abc|xyz) = P(abc|z)\ \forall\ a, b, c, x, y, z$. We derive this by running again a vertex enumeration for this scenario and checking that this holds for all extremal vertices. Indeed, only a few extremal correlations are left in this case, which we list in Supplementary Note 2, all of which satisfy (5) and (6). This implies that there is a monogamy relation between these relativistically causal influences, where the effect of $Z$ on $A$, $B$ restricts possible influences of the space-like separated $X$ and $Y$.

We remark here that monogamy relations between the influences of $X, Y, Z$ are not restricted to this particular example but hold more generally; the example is chosen for ease of presentation. In Supplementary Note 2, we also show monogamy for other functional dependencies of the variables in this setup. Specifically, we consider the setup of Fig. 2 where each of the six parties chooses inputs and obtains outcomes. We then show that if correlations previously considered in ref. 8 are established in a sub-scenario of the setup, they imply monogamy relations in the sense that they prevent other parties from achieving the same dependencies.

We further derive general constraints on the information the variables $X, Y, Z$ can hold about the pairs of variables $(B, C), (A, C), (A, B)$

respectively in the setup of Fig. 2. A natural measure for capturing this is the mutual information of two random variables $S$, $T$, $I(S: T) = H(S) + H(T) - H(ST)$, where $H(S) = -\sum_s P(s)\log P(s)$ is the Shannon entropy of the random variable $S$. The mutual information is non-negative and it is zero iff $P(st) = P(s)P(t)\ \forall\ s, t$, thus indeed capturing the information $T$ holds about $S$ and vice versa. We will further use the conditional mutual information of two random variables $S$, $T$ conditioned on a third, $V$, defined as $I(S: T|V) = H(SV) + H(TV) - H(V) - H(STV)$, which is well-known to be non-negative for any sets of random variables $S$, $T$, $V$ and zero iff $P(st|v) = P(s|v)P(t|v)\ \forall\ s, t, v$.

Let us first consider random variables $A, B, C, X, Y, Z$ with restricted cardinality, again in the setup of Fig. 2. For $X, Y, Z$ uniformly random bits, and binary $A, B, C$, we find that

$$I(AB : Z) + I(AC : Y) + I(BC : X) \leq 1. \tag{7}$$

This is a non-trivial restriction on the $P(abcxyz)$, as each of the terms $I(AB: Z)$, $I(AC: Y)$, $I(BC: X)$ can separately take the value 1. To prove the inequality, we can check that it holds for all of the 11964 extremal vertices of $\mathcal{K}$ (see above). Furthermore, $P(x)$, $P(y)$, $P(z)$ respectively are fixed in this case and we know that the mutual information is a convex function in $P(bc|x)$, $P(ac|y)$, $P(ab|z)$ respectively, which concludes the proof since the convex combination of two distributions $P_1$, $P_2$, $\lambda P_1(abcxyz) + (1 - \lambda)P_2(abcxyz)$ in this case corresponds to a convex combination $\lambda P_1(ab|z) + (1 - \lambda)P_2(ab|z)$, $\lambda P_1(ac|y) + (1 - \lambda)P_2(ac|y)$, $\lambda P_1(bc|x) + (1 - \lambda)P_2(bc|x)$ and a convex function on a convex set takes its maximal value at an extreme point, which in a polytope is at a vertex.

This entropy inequality captures the monogamy between the influences of $X, Y, Z$: it ensures that if $Z$ has maximal information about $A, B$, which in this case means $I(Z: AB) = 1$, then $I(Y: AC) = I(X: BC) = 0$, which is the case if and only if $Y$ is independent of $AC$ and $X$ is independent of $AB$. The strategy $Z = A \oplus B$ above is an example that saturates this inequality.

We can further derive constraints that hold independently of the cardinality of the involved variables (and without assuming uniform $X, Y, Z$):

$$I(X : BC) + I(Y : AC) \leq H(C|AB) \tag{8}$$

and permutations of the triples $(X, A, B)$, $(Y, C, A)$, $(Z, B, C)$. To prove this inequality, let's take the sum of the following non-negative entropic quantities: $I(X: Y|ABCZ)$, $I(Y: Z|ABC)$, $I(X: Z|ABC)$, $I(X: A|BC)$, $I(Y: B|AC)$, $I(Z: C|AB)$, $H(ABCXYZ) - H(X) - H(Y) - H(ABZ)$. Rewriting this sum leads to (8). Note that non-negativity of the last quantity follows from

$H(C|ABXYZ) \geq 0$ and $H(ABXYZ) = H(ABZ) + H(X) + H(Y)$, which must hold for any distribution in this configuration where the $X, Y, Z$ are chosen independently (due to the relativistic causality constraint $P(ab|xyz) = P(ab|z)\ \forall a, b, x, y, z$). For a systematic method to derive further such entropy inequalities, we refer to the "Methods".

Intuitively, (8) captures that if $X$ and $Y$ hold a lot of information about correlations of the pairs $B, C$ and $A, C$ respectively, then this restricts the possible dependency of $C$ on $AB$ (which is highest if $H(C|AB) = 0$). Intuitively, this has to be the case since otherwise there would be information about $X$ and $Y$ also contained in $C$, which is not allowed in this setup.

While the triangular arrangement of Fig. 2 exhibits an appealing symmetry, an immediate question is whether this type of relation also arises in a smaller setup. Considering just one of the variables $X, Y, Z$ cannot lead to similar relations, however, dropping only one of them can lead to similar effects. We treat such an example in detail in Supplementary Note 2, where we classify all correlations. There, we furthermore provide examples of correlations where both $X$ and $Y$ have an effect on the correlations of $A, B, C$ (but in a weaker form due to the monogamy relations). We further identify cases where $X$ and $Y$ only jointly affect the correlations of $A, B, C$.

## Consequences of jamming

Our monogamy relations show that relativistically causal correlations (and thus also potential mechanisms leading to them) can be highly non-local. Here, we illustrate this with an example. We show that while the correlations themselves do not (by definition) feature a superluminal transfer of information, inconsistencies that could be exploited to send superluminal signals can occur if we assume that there is a mechanism that an agent can voluntarily turn on and off that affects these relativistically causal effects. We call this a jamming mechanism[7]. Specifically, in the following, we show that we cannot have the two following features at the same time: (i) a general jamming mechanism that parties can turn on and off (ii) relativistically causal correlations P of $A, B, C, X, Y$.

To see this, let us consider a sub-scenario of the triangular setup above, see Fig. 3a. We consider three parties, Alice, Bob and Charlie, who always input 0 into their devices and get outcomes $a$, $b$, $c$, respectively, and Xavier and Yanina, who choose $(x, x_m)$ and $(y, y_m)$ respectively and obtain an output 0 (where the deterministic inputs and outcomes 0 are not shown in the figure).

Now assume that the jamming mechanism that Xavier and Yanina can trigger causes $X$ to influence $A, B$ and $Y$ to influence $B, C$. This is analogous to the scheme in ref. 7, where a party (e.g., Xavier here) can turn on a jamming device that jams the correlations of two others.

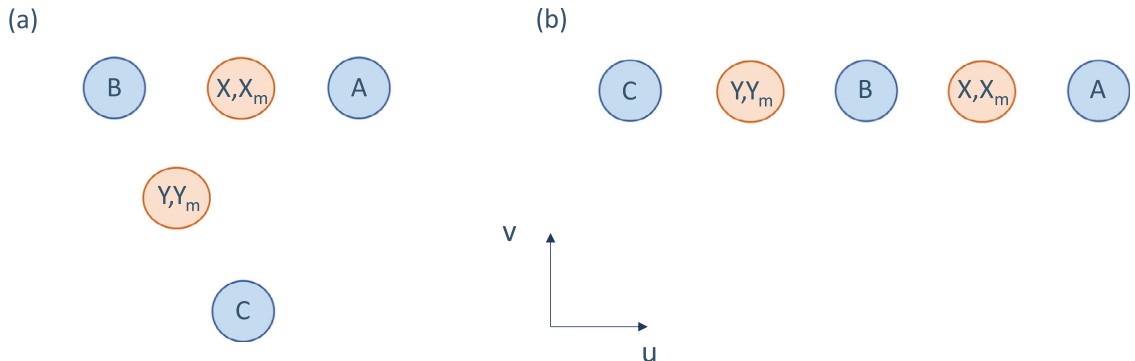

**Fig. 3 | Spatial arrangements of five parties. a** Compass setup. The two parties Xavier and Yanina (orange) choose inputs $(X, X_m)$, $(Y, Y_m)$ while Alice, Bob and Charlie (blue) generate outputs $A$, $B$ and $C$ respectively. The five events are all space-like separated and are displayed here in a frame where they are simultaneous. **b** Line setup. This scenario, compared to the compass, does not feature the constraint $P(ac|xy) = P(ac)\ \forall\ a, c, x, y$, while the others remain the same.

Now, first of all, assume that Xavier's jamming mechanism causes the outputs of Alice and Bob to satisfy $A \oplus B = X$, while without it, $A$ and $B$ are independent. Such a mechanism leads to valid relativistically causal correlations in the sub-scenario that includes $A$, $B$, $X$ (as e.g., considered in Fig. 1), which is straightforward to check. If Xavier can turn this mechanism on and off (see (i)), he can encode the decision to turn his jamming mechanism on as $X_m = 1$ or off as $X_m = 0$. The same strategy can be followed by Yanina for $B$, $C$, $Y$ and $Y_m$.

The scenario considered here is a sub-scenario of the triangle considered above, where we drop one of the sources. In this case, for binary variables $A$, $B$, $C$, we are left with the monogamy relation $I(X{:}AB) + I(Y{:}BC) \leq 1$. Hence, for relativistically causal correlations in this setup, whenever $A \oplus B = X$, $B$ and $C$ are independent (and vice versa). Thus the independent control of the jamming mechanisms (which allows for $X_m = Y_m = 1$) is inconsistent with relativistically causal distribution $P$ (see (ii)).

This means that for the distribution $P$ to be consistent with relativistic causality whenever Yanina chooses $Y_m = 1$, Xavier cannot choose $X_m = 1$. Thus by choosing $Y_m$ Yanina can restrict Xavier's choice: If $Y_m = 1$, only $X_m = 0$ is possible, if $Y_m = 0$ then $X_m \in \{0, 1\}$. The other way around, if we assume that Xavier and Yanina have control over a general jamming mechanism each, then the overall distribution $P$ will be outside of the relativistically causal set and thus signalling. Thus such a controlled jamming mechanism could be used to signal. The simplest way to see this is to consider the case where $X_m = Y_m = 1$, which implies that $X = Y \oplus A \oplus C$. Thus, Xavier can signal to Alice, Charlie and Yanina, who can use $A$, $C$, $Y$ to retrieve Xavier's bit $X$ outside of his future light cone [Note that there are signalling distributions that are consistent with the two jamming mechanisms: for each pair $X_m = x_m, Y_m = y_m$ we can construct some distribution $P$. For example, for $X_m = 1$, $Y_m = 1$ we can consider the distribution that has $P(000|00) = P(111|00) = P(010|11) = P(101|11) = P(001|01) = P(110|01) = P(011|10) = P(100|10) = 1/2$ and zero otherwise].

Now let us assume that the jamming mechanism is only partially controlled, namely, with probability $p > 0$ the mechanism of Yanina fails, i.e., if $Y_m = 1$, $P(B \oplus C = Y) = 1 - \frac{1}{2}p$ and $P(B \oplus C \neq Y) = \frac{1}{2}p$, otherwise $B$, $C$ are independent (and similarly for Xavier). This corresponds to a situation where the parties can only influence such a mechanism. Then if $Y_m = 1$, this implies that $P(A \oplus B = X) \leq \frac{1}{2}(1 - p) + p$. (If $Y_m = 0$, then $P(A \oplus B = X) \leq 1$.) Now if Xavier always chooses $X_m = 1$, then $P(A \oplus B = X) = 1 - \frac{1}{2}p$. This implies that $p >= 1/2$.

Thus, turning on (or not) a jamming device (e.g., $Y_m$) can, in this scenario, not strongly affect whether jamming occurs (e.g., $Y = B \oplus C$). Instead, this could potentially be achieved by some non-local mechanism that takes all parties that may attempt to jam into account. We discuss this by asking, from which point on information about a jamming effect could even exist. We illustrate this with the example of the influence of $X$ and/or $Y$ on $B$.

First, relativistically causal influences might take shape where $B$ is generated (or at one specific point in the intersection of the future light cones of $X$ and $Y$ before that). However, if a superluminal influence from $X$ and $Y$ exists only from $B$ on (and similarly for $A$, $C$), there needs to be a feature in that explanation that causes the scenarios in Fig. 3 to allow different correlations. Thus (non-local) information related to the positions of the other parties would need to be available at $B$ as well.

Second, whether $B$ is affected by the superluminal influences of $X$ and/or $Y$ might be determined where the future light cones of $X$ and $Y$ intersect. However, since for more than one spatial dimension the intersection of the future light cones of $X$ and $Y$ is not a light cone and does thus not have a single point where the information can start to exist, this would be a delocalised region. This means that along this

whole region, there would be a (non-local) decision whether the effect of $X$ or $Y$ should prevail.

Third, the jamming variables $X$, $Y$ might be (globally) pre-determined, so that it is already decided which one is to affect $A$, $B$, $C$ when they are generated. However, this is in tension with the relativistic causality framework, which relies on the premise that there are independent variables.

Any mechanism leading to relativistically causal correlations has to be highly non-local in a stronger sense than is observed from monogamy relations in quantum theory, e.g., for entanglement[21] or non-locality[22]. In the case of entanglement and non-locality, the monogamy concerns measurement on a system that is shared with several parties that measure it in the future, in the present type of monogamy it concerns the influences of variables ($X$, $Y$) that are independent and space-like separated.

Causal reasoning in the Bayesian sense[23] has been successfully adapted from its classical formulation for random variables to the quantum realm[24,25], where notions such as d-separation and the notion of interventions were formulated for quantum processes. Since quantum theory may one day be replaced by a more general theory (hopefully including gravity), a more general, theory-independent framework to study causal relations has been developed for generalised probabilistic theories[26]—which (as mentioned above) satisfy all non-signalling constraints[5]—as well as more generally[11]. This latter framework is, in particular, able to incorporate causal influences as considered in this work, allowing for fine-tuned causal relations[11]. The monogamy relations considered here, however, provide a conceptual problem for causal reasoning. Causal modelling presupposes that there is an underlying mechanism for correlations. Our work suggests that such mechanisms have likely a global nature and can not be (partially) controlled by agents. This questions whether causal modelling makes sense for such correlations in setups in more than one spatial dimension, i.e., whether ref. 11 can be generalised to this regime.

Furthermore, by looking at a sub-scenario of the one above, say $A$, $B$, $X$, we can potentially not know whether the variables have to follow the non-signalling constraints (see Supplementary Equation (S1)) or whether they are only restricted by (2), which, according to the monogamy relations, depends on other variables (in this case e.g., $Y$). Recall that, since we can consider additional inputs to $A$, $B$ and outcomes to $X$, this reasoning also applies to the original scenario of Fig. 1. This makes causal and also experimental reasoning about setups that allow for general relativistically causal correlations challenging, in the sense that we have to be able to exclude that there exist any variables (in principle anywhere in the universe) that may affect correlations e.g., of $A$ or $B$ with other variables. Thus considering such relativistically causal correlations would pose new challenges to the way we currently do research by considering only an experimental setup of interest and some environmental interactions, but where we can assume that no unknown space-like separated variables influence how we can operate our experiment.

## Further applications of entropy inequalities for relativistic causality

Entropy inequalities further give us a method to distinguish properties of non-signalling from relativistically causal correlations. We show in this section that the latter correlations still satisfy non-trivial constraints that are albeit weaker than those for correlations satisfying the full non-signalling constraints, which were derived in ref. 12.

That we can distinguish the correlations of different theories (e.g., classical and quantum) in networks with entropy inequalities, has only ever been shown for entropies where we condition on the inputs

taking specific values[27-29]. This motivates us to also pursue such an approach for relativistically causal correlations. Our method to obtain constraints for relativistically causal correlations is similar to the method for constraining full non-signalling correlations in ref. 12, to which we compare our constraints. We describe this in detail in the "Methods".

To illustrate the technique we show how conditions on the monogamy of non-locality in the scenario of Fig. 1 have to be relaxed in the relativistically causal case. Let us stress here that this is a conceptually different type of monogamy relation from those derived in the subsection 'Monogamy of relativistically causal correlations', the fact that both are known as monogamy relations is coincidental. Previously, it was shown in ref. 8 that the monogamy of the CHSH violation of Alice and Bob and Bob and Charlie is violated by some relativistically causal correlations. Here we address this same problem from the perspective of entropy inequalities, where we show how such inequalities are relaxed as compared to the full non-signalling setting.

For this purpose, let us consider the entropy of the distributions $P(ABC|X = x, Y = y, Z = z)$, where $H(A_x B_y C_z) = H(P(ABC|X = x, Y = y, Z = z))$ is the Shannon entropy of the conditional distribution. Now let us define

$$H_{AB}^{CHSH} := H(B_0|A_0) + H(A_1|B_0) + H(A_0|B_1) - H(A_1|B_1), \quad (9)$$

where $H(S|T) = H(ST) - H(T)$ is the conditional entropy of the random variable $S$ conditioned on $T$ (according to ref. 27, $H_{AB}^{CHSH}$ is positive for local models in the Bell scenario). Now, according to ref. 12, in the setting of three parties on a line,

$$H_{AB}^{CHSH} + H_{BC}^{CHSH} \geq 0, \quad (10)$$

which is an entropic version of the monogamy of the non-local correlations Bob holds with Alice and Charlie respectively. In the full non-signalling case this also holds for all permutations of the three parties and relabellings of inputs.

Following our technique detailed in the "Methods", we find that when Bob's input can affect the correlations of Alice and Charlie, then the entropic monogamy relation (10) breaks down and we are unable to derive a weaker form of it. However, for the correlations Alice shares with Bob and with Charlie there is still a non-trivial, if weaker, monogamy relation:

$$H_{AB}^{CHSH} + H(C_0|A_0 Y = 1) + H(A_1|C_0 Y = 1) \\ + H(A_0|C_1 Y = 0) - H(A_1|C_1 Y = 0) \geq 0, \quad (11)$$

where the second to fifth term reduce to $H_{AC}^{CHSH}$ when $Y$ does not affect the correlations of $A$ and $C$. For a general technique to derive entropy inequalities for relativistically causal correlations and the derivation of (11), we refer to the "Methods". Note also that in this setting the role of the two parties attempting to implement a protocol (here Alice and Bob) is not symmetric anymore.

## Discussion

This work shows that relativistically causal correlations have to satisfy strong monogamy relations, which come about when such correlations are considered in setups in more than one spatial dimension. Such multipartite scenarios arguably have to be analyzed if one is to take relativistically causal correlations seriously.

This calls into question the explanation of these correlations via a jamming mechanism in terms of a device that an agent can voluntarily turn on, as such a device could be used to signal. This is problematic in any single reference frame where an explanation of these correlations

may be attempted, not only when changing reference frames (which is known to lead to difficulties for explaining this type of correlation[8]). The monogamy relations are also problematic for the way we perform experiments, where we can usually assume that systems at far away space-time locations do not influence the correlations of our outcomes.

This lack of control about a jamming mechanism also implies that it is not so obvious what the implications of relativistically causal correlations might be for cryptography since it is not clear to what extent anyone could actively make use of them. The monogamy relations further imply that, if there exists a mechanism that consistently jams the correlations of some parties, this may shield other parties from similar interferences, which may serve as an asset for salvaging some cryptographic protocols (of which several have already been shown to be compromised in this framework in refs. 8,9).

## Methods
### A method for deriving entropy inequalities for relativistically causal correlations

Entropy allows us to formulate natural measures for quantifying the information some random variables hold about others, in terms of (conditional) mutual information. They furthermore have the convenient property that they linearise (conditional) independence constraints, which has made them one of the main tools to formulate causal compatibility constraints in networks in the past[23].

In a similar vein, any relativistic causality constraint according to Definition 1, leads also to an entropic constraint on $P(a_1, \ldots, a_n, x_1, \ldots, x_n)$, namely

$$H(\tilde{\mathcal{A}}_{\mathcal{J}} \tilde{\mathcal{X}}_{\mathcal{J}} \tilde{\mathcal{X}}_{\mathcal{J}^c}) = H(\tilde{\mathcal{A}}_{\mathcal{J}} \tilde{\mathcal{X}}_{\mathcal{J}} \tilde{\mathcal{X}}_{\mathcal{K}}) + H(\tilde{\mathcal{X}}_{\mathcal{J}^c} \setminus \tilde{\mathcal{X}}_{\mathcal{K}}), \quad (12)$$

where $\tilde{\mathcal{A}}_{\mathcal{J}} = \{A_i | i \in \mathcal{J}\}$ and $\tilde{\mathcal{X}}_{\mathcal{J}} = \{X_i | i \in \mathcal{J}\}$ and analogously for $\tilde{\mathcal{X}}_{\mathcal{K}}, \tilde{\mathcal{X}}_{\mathcal{J}^c}$. This can be concisely written as $I(\tilde{\mathcal{A}}_{\mathcal{J}} \tilde{\mathcal{X}}_{\mathcal{J}} \tilde{\mathcal{X}}_{\mathcal{K}} : \tilde{\mathcal{X}}_{\mathcal{J}^c} \setminus \tilde{\mathcal{X}}_{\mathcal{K}}) = 0$. It is well known that this holds iff $P(\tilde{\mathcal{A}}_{\mathcal{J}} \tilde{\mathcal{X}}_{\mathcal{J}} \tilde{\mathcal{X}}_{\mathcal{J}^c}) = P(\tilde{\mathcal{A}}_{\mathcal{J}} \tilde{\mathcal{X}}_{\mathcal{J}} \tilde{\mathcal{X}}_{\mathcal{K}}) P(\tilde{\mathcal{X}}_{\mathcal{J}^c} \setminus \tilde{\mathcal{X}}_{\mathcal{K}})$. Thus, these entropic relations fully capture the relativistic causality constraints. Recall that as inputs the $X_i$ are furthermore all independent, i.e., $H(X_1, \ldots X_n) = H(X_1) + \cdots + H(X_n)$.

All such constraints, together with the so-called Shannon inequalities

$$H(\mathcal{T}) \geq 0 \\ H(\mathcal{V}|\mathcal{W}) \geq 0 \\ I(\mathcal{T} : \mathcal{V}|\mathcal{W}) \geq 0 \quad (13)$$

for all non-intersecting sets of variables $\mathcal{T}, \mathcal{V}, \mathcal{W} \subset \{A_1, \ldots, A_n, X_1, \ldots, X_n\}$, form the so-called Shannon cone of the scenario

$$\mathcal{S} = \left\{ \mathbf{H} \in \mathbb{R}^{2^{2n}-1} | \text{(13) and (12) hold} \right\}, \quad (14)$$

where $\mathbf{H} = (H(A_1), \ldots, H(X_n), H(A_1 A_2), \ldots, H(A_1 \cdots A_n X_1 \cdots X_n))$.

Asking what these constraints impose on the relation of specific marginals can then be phrased as a variable elimination problem. Indeed, we can identify components of $\mathbf{H}$ that are of interest to us and then derive constraints only involving these variables by means of performing a Fourier-Motzkin elimination of all other variables on $\mathcal{S}$. We illustrate this technique with an example.

**Example 1.** Let us consider the triangle setup of Fig. 2. We consider the entropy vectors

$$\mathbf{H} = (H(A), H(B), \ldots, H(Z), H(AB), \ldots, H(ABCXYZ)), \quad (15)$$

for which we impose all Shannon inequalities as well as the following independence constraints:

$$H(ABXYZ) = H(ABZ) + H(X) + H(Y),$$
$$H(ACXYZ) = H(ACY) + H(X) + H(Z),$$
$$H(BCXYZ) = H(BCX) + H(Y) + H(Z),$$
$$H(AXYZ) = H(A) + H(X) + H(Y) + H(Z),$$
$$H(BXYZ) = H(B) + H(X) + H(Y) + H(Z),$$
$$H(CXYZ) = H(C) + H(X) + H(Y) + H(Z),$$
$$H(ACYZ) = H(ACY) + H(Z),$$
$$H(ACXY) = H(ACY) + H(X),  \quad (16)$$
$$H(ACXZ) = H(AC) + H(X) + H(Z),$$
$$H(ABYZ) = H(ABZ) + H(Y),$$
$$H(ABXZ) = H(ABZ) + H(X),$$
$$H(ABXY) = H(AB) + H(X) + H(Y),$$
$$H(BCXY) = H(BCX) + H(Y),$$
$$H(BCXZ) = H(BCX) + H(Z),$$
$$H(BCYZ) = H(BC) + H(Y) + H(Z).$$

In the context of monogamy relations, we are interested in constraints that only involve $H(ABZ)$, $H(ACY)$, $H(BCX)$ and their marginals. Thus we remove all other variables from the inequalities by means of a Fourier-Motzkin elimination, which we perform using the software PORTA[20]. From this elimination, we find six equality constraints:

$$
\begin{aligned}
H(AY) &= H(A) + H(Y), & H(AZ) &= H(A) + H(Z) \\
H(BX) &= H(B) + H(X), & H(BZ) &= H(B) + H(Z)  \quad (17) \\
H(CX) &= H(C) + H(X), & H(CY) &= H(C) + H(Y)
\end{aligned}
$$

and one interesting class of inequality, namely,

$$I(X : BC) + I(Y : AC) \leq \min\{H(C|B), H(C|A)\}, \quad (18)$$

and permutations of the triples $(X, B, C)$, $(Y, C, A)$, $(Z, A, B)$. All other inequalities are Shannon inequalities. [Notice that this differs slightly from (8), as in the example presented here for illustration, we are interested in relaxed constraints that do not involve $H(ABC)$.].

We remark that bounds on the cardinality $|S|$ of any involved variable $S$, can in principle be incorporated in this method by imposing additional upper bounds

$$H(S) \leq \log |S|. \quad (19)$$

This is a relaxation of an actual restriction of the cardinality, as there are variables with higher cardinality that also satisfy the inequality. In this scenario, this did not lead to any interesting inequalities.

Entropy inequalities can further be formulated where we condition on some of the variables, usually the inputs, to take specific values. Such methods have been called post-selection techniques[29]. Similar ideas have been applied to analysing networks without the use of entropy, where conditioning on certain variables taking specific values is now known as unpacking[30]. The method introduced here is inspired by ref. 12 and allows us to directly compare the inequalities for relativistically causal correlations to those arising from the full non-signalling correlations from ref. 12.

For this, let us consider a collection of pairs of (input, output) variables $(X_i, A_i)$ that can take values $x_i \in \mathcal{X}_i$ and $a_i \in \mathcal{A}_i$ respectively. Then let us consider a network involving all of these variables with a directed edge from input to outcome within each pair. Now, for each $X_i$ in the network, we identify all variables $A_j$ of which $X_i$ may influence the correlations with some other variables and we add directed edges from $X_i$ to all of these variables. After this, we consider each $A_j$ and we create a copy of this variable for each combination of values the variables influencing it can take. For instance, if $A_j$ is directly influenced in this network by $X_j, X_k, X_l$ (meaning that there is a directed edge pointing to it from those variables), then we create $|\mathcal{X}_j| \cdot |\mathcal{X}_k| \cdot |\mathcal{X}_l|$ copies of $A_j$, indexed by the values $x_j, x_k, x_l$, namely $A_j(x_j, x_k, x_l)$. These represent random variables with probabilities $P(a_j|x_j, x_k, x_l)$.

Now among all the variables $\{A_i(\cdot)\}_i$ we define sets of coexisting variables, which are those that agree in the value that the $X_j$ that they share as an index take and we drop all such sets that are subsets of others, thus leaving us with maximal coexisting sets. These are the maximal sets of variables for which we can define joint conditional distributions. For each of these, we build an entropy vector.

Now, each of these entropy vectors has to satisfy all Shannon inequalities (see above). In addition, the relativistic causality constraints impose relations among the entropies: If a variable, $X_j$, affects only the correlations of another, $A_i$, the entropy of $A_i$ will not depend on $X_j$, i.e., $H(A_i(x)) = H(A_i(x'))$ for two values $x, x'$ of $X_j$. Analogous relations may arise for sets of variables $\{A_i\}_i$. Note furthermore that the entropy vectors share some of the variables, so that we obtain an overall system of linear entropy inequalities by combining the inequalities from all the maximal coexisting sets.

Now we can use Fourier-Motzkin elimination to obtain constraints that only involve the variables we are interested in. We illustrate this whole procedure with an example.

**Example 2.** Let us consider the scenario from Fig. 1, with binary inputs $X$, $Y$, $Z$. This means that we define 10 variables $A_{00} = A_{x=0,y=0}$, $A_{01} = A_{x=0,y=1}$, $A_{10} = A_{x=1,y=0}$, $A_{11} = A_{x=1,y=1}$, $B_0 = B_{y=0}$, $B_1 = B_{y=1}$, $C_{00} = C_{y=0,z=0}$, $C_{01} = C_{y=0,z=1}$, $C_{10} = C_{y=1,z=0}$, $C_{11} = C_{y=1,z=1}$. These form six maximal coexisting sets $\{A_{x_0 y_0}, B_{y_0}, C_{y_0 z_0}\}$ for $x_0, y_0, z_0 \in \{0, 1\}$. For each set, we obtain a vector

$$\mathbf{H}_{x_0, y_0, z_0} = (H(A_{x_0 y_0}), H(B_{y_0}), \ldots, H(A_{x_0 y_0} B_{y_0} C_{y_0 z_0})). \quad (20)$$

In this case, relativistic causality imposes that

$$
\begin{aligned}
H(A_{00}) &= H(A_{01}), & H(A_{10}) &= H(A_{11}) \\
H(C_{00}) &= H(C_{10}), & H(C_{01}) &= H(A_{11}).
\end{aligned} \quad (21)
$$

We further impose all Shannon inequalities.

In our example we are interested in the tradeoff between the CHSH violation of Alice and Bob, Alice and Charlie. We thus aim to derive constraints involving $H_{AB}^{\text{CHSH}}$, $H_{AC}^{\text{CHSH}}$, (see (9) in the main text). For this we remove entropies involving three of the variables as well as those involving $B$ and $C$ from the scenario. Performing this variable elimination in PORTA[20], we obtain Shannon inequalities and 16 inequalities of the type (11).

## Data availability
All relevant data is contained in the main text and the Supplementary Information. Any additional calculations can be obtained from the author upon reasonable request.

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

## Acknowledgements
The author would like to thank Paul Skrzypczyk and Giorgos Eftaxias for helpful discussions and Roger Colbeck and Peter Brown for feedback on a draft of this work. This work was funded by the Swiss National Science Foundation (Ambizione PZ00P2_208779).

## Author contributions

M.W. completed this work.

## Competing interests
The author declares no competing interests.
