## [Transparent Peer Review file · Nature Communications]

Monogamy relations for relativistically causal correlations

Corresponding Author: Dr Mirjam Weilenmann

Version 0:

Reviewer comments:

Reviewer #1

(Remarks to the Author)

In this work, the exploration of relativistically causal (RC) correlations in multi-party quantum systems reveals unexpected higher degrees of non-locality, suggesting the presence of signaling, a concept traditionally deemed incompatible with this theory. While the paper is well-written and organized, the explanation of how signaling occurs within these constraints is not entirely clear. I recommend that the authors refine their explanation, making the argument more concise, intuitive and accessible. This would greatly aid in understanding the significant implications of these findings.

Considering the feedback provided, I suggest a major revision of the work's primary claim. This revision should aim to address the highlighted issues before resubmitting to Nature Communications. The major revision should focus on clarifying and substantiating the argument, especially regarding the presence of signaling in RC correlations, to align with the journal's standards and expectations.

Kind regards,

The Referee

Reviewer #2

(Remarks to the Author)

The manuscript investigates the consequences of relaxing non-signaling conditions in multipartite scenarios in terms of the so-called relativistically causal correlations (RCC) introduced by Horodecki & Ramanathan (Nat. Comm. 10, 2019). The author derives entropic monogamy relations that must be satisfied by such theories and show how they imply that previous suggestions of a physical mechanisms at the basis of RCC actually implies superluminal signaling, so it must be refuted.

The main motivation for the introduction of RCC (Horodecki & Ramanathan, Nat. Comm. 10, 2019, Ref. 8) was the observation that non-signaling (NS) conditions are not necessary to enforce relativistic causality, i.e., avoiding causal loops, in some specific scenarios. Mathematically, one relaxes the NS conditions by allowing one variable, X , to influence other variables, $(A_i)_i$, if the intersection of the future light cones for $(A_i)_i$ lies fully in the future light cone of X . For instance, in the case of three parties in a line, say A, B, C , it is possible for B to influence the correlation between A and C . As a physical mechanism at the basis of this phenomenon, a device that "jams" correlations of other parties was suggested in Ref. 8, recovering a proposal by Grunhaus, Popescu, and Rorhlich.

The present manuscript shows that, even if the correlations themselves do not feature a superluminal transfer of information, superluminal signaling can occur if one assumes that there is a mechanism that an agent can turn on, i.e., the jamming device, that leads to such relativistically causal effects. This is achieved by showing that RCCs obey some monogamy relations, which prevents the simultaneous jamming of more than one pair of the variable, in a 2D spatial configuration of the parties that extends the 1D example of Ref. 8. This implies that the jamming device could be used to signal faster than light

among the jamming parties. The result remains true even if the jamming parties have only a partial control of their device, i.e., with probability p .

In addition to that, the authors also derive other monogamy relations, for the case in which the output variables A, B, C (same spatial configuration) are conditioned on the input variables X, Y, Z .

The manuscript is very clear and well written. All the technical results seem correct, even though I didn't check the numerical results. However, since the latter are obtained with standard methods (FM algorithm), I am confident that they are correct. I don't have particular suggestions for editing the manuscript, with the exception of a couple of typo that I found: repetition of the set I twice above Eq. (C1), and Yanina is sometimes spelled Yannina, I guess both spellings are correct, one should just pick one.

The manuscript presents a very important result that strongly undermines the plausibility of the whole framework of relativistically causal correlations, which received substantial attention in the last few years. This, in turns, has implications on the security of cryptographic protocols that were questioned on the basis of such a framework.

Such results are definitely of interest for the broad audience of scientists interested in quantum information and technology. I strongly recommend the manuscript for publication.

Reviewer #3

(Remarks to the Author)

The paper deals with the characterisation of the set of "relativistically causal correlations" (RCC here below). It follows a series of works that stem from harnessing quantum correlations. There is a famous relaxation of quantum correlations, called no-signaling correlations, which have been studied intensely in the last two decades. In 2019, Ref [8] provided a less constrained relaxation, the RCC that are the object of this study. Obviously quantum correlations are RCC (since they obey the stricter constraint of no-signaling).

The rationale is nicely understood in Figure 2: in that space-time arrangement of the parties, A and C can share their information (x, a) and (z, c) only in the intersection of their respective light cones. That intersection is fully in the future of B . No-signaling prescribes that $P(a, c|x, y, z)$ does not depend on y . The idea of RCC is that one could let $P(a, c|x, y, z)$ depend on y , because A and C cannot profit of the dependence on y to communicate faster than light.

Clearly, these are foundational concerns. In terms of "applications", two motivations for such studies are usually given. First, since we don't have a quantum theory of gravity, it may well happen that quantum correlations are not the last word: one therefore studies larger sets of correlations that are constrained only by relativistic demands. Second, one may want to certify some devices at what I would call super-device-independent level (assuming no-signaling, this has been done for key distribution and randomness expansion).

The contributions of the paper are:

(1) An advanced study of constraints that RCC must satisfy. The results are novel to the best of my knowledge; they are correct, and the presentation can be easily followed.

(2) An analysis of the possible underlying physical mechanisms that could generate such correlations. This analysis is also correct, but I think it is already superseded by existing literature (see comment #6).

Here come my comments:

1. The introduction (Sections I and II) is too long. Leaving aside the formal definition (see comment #3), the few lines I have written above say it all. I encourage the author to shorten that part.

2. In the same vein, Figure 2 says it all very clearly and should have appeared very early. By contrast, Figure 1 seems useless.

3. The formal definition II.1 is not easy to read. First of all, rather than forcing the reader to remember that J is the complement of I , I would define only J and write J^c in the sum (thus dispensing from defining I). Then, instead of defining K as a generic superset of J , whose meaning is taken up only by the remark "where we omit...", I suggest a more striking writing: define K as those x 's in J^c that are retained, and write the rhs as $P(A|X_J, X_K)$. This way, it becomes visually obvious that RCC is a relaxation of no-signaling.

4. Section III starts by talking casually of an example (that of Figure 3): it gave me the impression that it was just an introductory moment. But in reality, all the technical content presented in the main text is about that example. I suggest rather writing: "in the main text, we illustrate our main results by studying the example of Figure 3; generalisations are left for the appendices". Then the reader knows what to expect.

5. On the same discussion of figure 3: I had some problems counting "parties", till I realized that those who choose the inputs are also considered parties. I see why this is needed here, and it would be silly to argue on wording. But it may be worth stressing that a CHSH test is a three-party scenario in this language, as most of the works on Bell nonlocality call it a two-party game.

6. On the "underlying explanations" (end of Section II, then Section IV): we do not have such underlying mechanisms even for quantum correlations; most of the community is not looking for them, because it is known that, if you look, you must find something "strongly nonlocal". Specifically, consider the series of works started in <https://arxiv.org/abs/quant-ph/0110074> (also with three parties and spacetime locations), and culminating in the proofs that any underlying mechanism should involve influences at infinite speed, first proved for no-signaling (<https://arxiv.org/abs/1102.5685>) and then for quantum correlations (<https://arxiv.org/abs/1110.3795>, <https://arxiv.org/abs/1304.1812>, <https://arxiv.org/abs/1304.0532>). Since such an extreme statement (frame-dependent of course, as duly noticed in the current paper) exists already for quantum and no-signaling, that are subsets of RCC, it seems to me that RCC cannot fare any better. If I am right, all those sections could be

replaced in the main text by a short statement.

7. As a follow up on #6: the criticism of jamming (currently core of Section IV) may be left in an Appendix for the sake of the history of foundations. Notice that x_m and y_m are undefined when they appear, only later I think I understand what their role is. In the example given, although measurement dependence and signaling are very related (see e.g. <https://arxiv.org/abs/2105.09037>), I would not say that "Yanina can signal to Xavier". The correct sentence appears later, the parties cannot choose their inputs individually (more strikingly: the parties have no free will).

Version 1:

Reviewer comments:

Reviewer #1

(Remarks to the Author)

I would like to thank to the author for sending a clarified explanation about signaling, introduced in Section II B. The fact that a general jamming mechanism, that parties can turn on/off, is incompatible with relativistic causal correlations is a surprising result, at least to me. Now that I understand how does it work, I believe it deserves to be published in Nature Communications. The new version clarifies the work in many aspects and I believe it is now understandable for experts and comprehensible, in general, for a broad audience.

Based on the above comments, I recommend the Manuscript NCOMMS-23-58174A for Publication in Nature Communications, as it is.

Reviewer #3

(Remarks to the Author)

The author has convincingly addressed my concerns. In the current version of the paper, the claims are clearer. I still find the discussion long and wordy (I didn't mention it before because I had suggested to move it to the Appendix). I leave it to the author whether it is possible to streamline that part.

As a footnote, I would still contend that we don't have an "explanation" even for quantum correlations. The theory we do have provides causality in Hilbert space, but not in space-time. Any attempt to bring those considerations into space-time has met with failure so far. Anyway, just for the record.

Resubmission of the manuscript

Monogamy relations for relativistically causal correlations

Point by point response to reviewers

I thank all reviewers for carefully reading my submission and for their feedback. In the following I respond to the comments by each of the reviewers separately.

Reply to Reviewer 1

In this work, the exploration of relativistically causal (RC) correlations in multi-party quantum systems reveals unexpected higher degrees of non-locality, suggesting the presence of signaling, a concept traditionally deemed incompatible with this theory. While the paper is well-written and organized, the explanation of how signaling occurs within these constraints is not entirely clear. I recommend that the authors refine their explanation, making the argument more concise, intuitive and accessible. This would greatly aid in understanding the significant implications of these findings.

Considering the feedback provided, I suggest a major revision of the work's primary claim. This revision should aim to address the highlighted issues before resubmitting to Nature Communications. The major revision should focus on clarifying and substantiating the argument, especially regarding the presence of signaling in RC correlations, to align with the journal's standards and expectations.

I thank the referee for feedback. I agree that the explanation of the signalling issue was not refined enough in the previous version and I have improved on this, see the changes in Section II B of the new version.

Essentially, the issue comes about when requiring that we have relativistically causal correlations (which by definition don't allow agents to signal) and require that agents can partially control whether a mechanism jams correlations. Section II B shows that the two are inconsistent. This means that either such a mechanism that the parties control cannot exist or that such a mechanism may exist but we have to allow superluminal signalling.

Reply to Reviewer 2

The manuscript investigates the consequences of relaxing non-signaling conditions in multipartite scenarios in terms of the so-called relativistically causal correlations (RCC) introduced by Horodecki & Ramanathan (Nat. Comm. 10, 2019). The author derives entropic monogamy relations that must be satisfied by such theories and show how they imply that previous suggestions of a physical mechanisms at the basis of RCC actually implies superluminal signaling, so it must be refuted.

The main motivation for the introduction of RCC (Horodecki & Ramanathan, Nat. Comm. 10, 2019, Ref. 8) was the observation that non-signaling (NS) conditions are not necessary to enforce relativistic causality, i.e., avoiding causal loops, in some specific scenarios. Mathematically, one relaxes the NS conditions by allowing one variable, X , to influence other variables, $(A_i)_i$, if the intersection of the future light cones for $(A_i)_i$ lies fully in the future light cone of X . For instance, in the case of three parties in a line, say A, B, C , it is possible for B to influence the correlation between A and C . As a physical mechanism at the basis of this phenomenon, a device that "jams" correlations of other parties was suggested in Ref. 8, recovering a proposal by Grunhaus, Popescu, and Rorhlich.

The present manuscript shows that, even if the correlations themselves do not feature a superluminal transfer of information, superluminal signaling can occur if one assumes that there is a mechanism that an agent can turn on, i.e., the jamming device, that leads to such relativistically causal effects. This is achieved by showing that RCCs obey some monogamy relations, which prevents the simultaneous jamming of more than one pair of the variable, in a 2D spatial configuration of the parties that extends the 1D example of Ref. 8. This implies that the jamming device could be used to signal faster than light

among the jamming parties. The result remains true even if the jamming parties have only a partial control of their device, i.e., with probability p .

In addition to that, the authors also derive other monogamy relations, for the case in which the output variables A, B, C (same spatial configuration) are conditioned on the input variables X, Y, Z .

The manuscript is very clear and well written. All the technical results seem correct, even though I didn't check the numerical results. However, since the latter are obtained with standard methods (FM algorithm), I am confident that they are correct. I don't have particular suggestions for editing the manuscript, with the exception of a couple of typo that I found: repetition of the set I twice above Eq. (C1), and Yanina is sometimes spelled Yannina, I guess both spellings are correct, one should just pick one.

I agree with the summary of the content of the manuscript and thank the referee for pointing out these errors, which are all corrected in the revised version.

The manuscript presents a very important result that strongly undermines the plausibility of the whole framework of relativistically causal correlations, which received substantial attention in the last few years. This, in turns, has implications on the security of cryptographic protocols that were questioned on the basis of such a framework.

Such results are definitely of interest for the broad audience of scientists interested in quantum information and technology. I strongly recommend the manuscript for publication.

I thank the referee for the positive assessment of the manuscript and for the recommendation.

Reply to Reviewer 3

The paper deals with the characterisation of the set of "relativistically causal correlations" (RCC here below). It follows a series of works that stem from harnessing quantum correlations. There is a famous relaxation of quantum correlations, called no-signaling correlations, which have been studied intensely in the last two decades. In 2019, Ref [8] provided a less constrained relaxation, the RCC that are the object of this study. Obviously quantum correlations are RCC (since they obey the stricter constraint of no-signaling). The rationale is nicely understood in Figure 2: in that space-time arrangement of the parties, A and C can share their information (x,a) and (z,c) only in the intersection of their respective light cones. That intersection is fully in the future of B. No-signaling prescribes that $P(a,c|x,y,z)$ does not depend on y . The idea of RCC is that one could let $P(a,c|x,y,z)$ depend on y , because A and C cannot profit of the dependence on y to communicate faster than light. Clearly, these are foundational concerns. In terms of "applications", two motivation for such studies are usually given. First, since we don't have a quantum theory of gravity, it may well happen that quantum correlations are not the last word: one therefore studies larger sets of correlations that are constrained only by relativistic demands. Second, one may want to certify some devices at what I would call super-device-independent level (assuming no-signaling, this has been done for key distribution and randomness expansion). The contributions of the paper are: (1) An advanced study of constraints that RCC must satisfy. The results are novel to the best of my knowledge; they are correct, and the presentation can be easily followed. (2) An analysis of the possible underlying physical mechanisms that could generate such correlations. This analysis is also correct, but I think it is already superseded by existing literature (see comment #6).

I thank the referee for the summary of the manuscript. I disagree, however, with the claim that the analysis of possible underlying mechanisms is superseded by the existing literature, see the responses to comments #5 and #6 for a detailed explanation.

Here come my comments:

1. The introduction (Sections I and II) is too long. Leaving aside the formal definition (see comment #3), the few lines I have written above say it all. I encourage the author to shorten that part.

I shortened both sections considerably. In particular, I removed the definition of the (multi-partite) non-signalling constraints, removed the explanation of the simpler setup with just one spatial dimension (as suggested in comment #2) and shortened the text in general.

2. In the same vein, Figure 2 says it all very clearly and should have appeared very early. By contrast, Figure 1 seems useless.

I have followed the request of the reviewer and removed Figure 1 from the manuscript (including the explanation of the simpler scenario when there is only one spatial dimension). Figure 2 furthermore appears earlier in Section II now. [Notice that Section II has been integrated into Section III in the new version.]

3. The formal definition II.1 is not easy to read. First of all, rather than forcing the reader to remember that J is the complement of I , I would define only J and write J^c in the sum (thus dispensing from defining I). Then, instead of defining K as a generic superset of J , whose meaning is taken up only by the remark "where we omit...", I suggest a more striking writing: define K as those $x \in J^c$ that are retained, and write the rhs as $P(AJ|XJ, XK)$. This way, it becomes visually obvious that RCC is a relaxation of no-signaling.

I agree, the issue is that there is a tradeoff between having a simple Definition II.1 and entropic independence constraints that look simple later. I rewrote Definition II.1 following the reviewer's advice.

4. Section III starts by talking casually of an example (that of Figure 3): it gave me the impression that it was just an introductory moment. But in reality, all the technical content presented in the main text is about that example. I suggest rather writing: "in the main text, we illustrate our main results by studying the example of Figure 3; generalisations are left for the appendices". Then the reader knows what to expect.

I clarified this at the beginning of Section III (now Section II A).

5. On the same discussion of figure 3: I had some problems counting "parties", till I realized that those who choose the inputs are also considered parties. I see why this is needed here, and it would be silly to argue on wording. But it may be worth stressing that a CHSH test is a three-party scenario in this language, as most of the works on Bell nonlocality call it a two-party game.

I respectfully disagree with this comment. The usual CHSH test is a two-party game also in this formulation. The scenario in Figure 3 (now Figure 2 in the new version) considers 6 space-like separated parties that can all choose inputs and obtain outcomes. Because it is easier and sufficient for observing the simplest monogamy relations, we choose Alice, Bob and Charlie to always choose input 0 and Xavier, Yanina and "Zorro" to always obtain output 0 (which is allowed in this setup).

One could consider turning the CHSH game into a 3-party game, for instance by considering preparations of different quantum states by a third party who prepares the different states depending on some input (and who may obtain an outcome from this as well). This will be a different arrangement though, because such a preparation has to occur before the respective states are measured, thus this party would not be space-like separated from Alice and Bob but in their past. In the setup here, all parties are spacelike separated.

In the manuscript I added another sub-figure to Figure 3 (now Figure 2 in the new version), that clarifies that all parties can have inputs and outcomes.

6. On the "underlying explanations" (end of Section II, then Section IV): we do not have such underlying mechanisms even for quantum correlations; most of the community is not looking for them, because it is known that, if you look, you must find something "strongly nonlocal". Specifically, consider the series of works started in <https://arxiv.org/abs/quant-ph/0110074> (also with three parties and spacetime locations), and culminating in the proofs that any underlying mechanism should involve influences at infinite speed, first proved for no-signaling (<https://arxiv.org/abs/1102.5685>) and then for quantum correlations (<https://arxiv.org/abs/1110.3795>, <https://arxiv.org/abs/1304.1812>, <https://arxiv.org/abs/1304.0532>). Since such an extreme statement (frame-dependent of course, as duly noticed in the current paper) exists already for quantum and no-signaling, that are subsets of RCC, it seems to me that RCC cannot fare any better. If I am right, all those sections could be replaced in the main text by a short statement.

I disagree with this comment. We do have an underlying explanation of quantum correlations. Namely, we know how to describe quantum correlations via the preparation of a quantum state that is measured by separated parties. This is a mechanism that explains all quantum correlations (even though not based on hidden influences in the sense of the papers above). For relativistically causal correlations we do not

have *any* underlying explanation. Since generalised probabilistic theories have non-signalling built into the framework, there is also no hope to find an explanation via generalised theories of this type for instance. In a scenario where we have relativistically causal correlations beyond non-signalling, we are thus missing any mechanism or explanation (whether classical, quantum or even more general). What is more, the correlations are such that there already is an influence from some variables to the correlations of others that are space-like separated from them, so there is a type of hidden influence built into the framework.

What a mechanism leading to relativistically causal correlations might look like is, as far as I am aware, completely unclear at this stage. One thing one could imagine is a hybrid of an explanation in terms of a generalised probabilistic theory supplemented with some hidden communication effects. This seems unnatural but would go beyond what is considered in hidden communication papers for instance. I added a comment outlining that we are not looking to replace all explanations of correlations by hidden communication in what was previously Section II and also cite all the papers the reviewer suggests there (this is now in Section II, just before subsection A starts).

Specifically concerning the reasoning in Section IV (now Section IIB in the new version): there are two aspects addressed there; firstly, whether there could be a mechanism, controlled by parties that turns relativistically causal influences on and off, and, secondly, where in spacetime the information about whether the variables (X, X_m) , (Y, X_m) superluminally influence the correlations of others can be determined and whether this can happen at a specific point in space-time. This second question is different from asking whether any relativistically causal correlations could be reproduced by hidden influences. I changed the phrasing at several places in this Section (specifically now in Section IIB1) to make these points clearer.

That the first result is different from results on hidden communication can be seen e.g. in that the argument in the current manuscript does not make a distinction between hidden influences at finite and infinite speed. The argument applies independently of the speed and in that sense goes beyond work on hidden communication, where the main point was to rule out hidden influences at finite speed.

7. As a follow up on #6: the criticism of jamming (currently core of Section IV) may be left in an Appendix for the sake of the history of foundations.

I have not moved this to the Appendix. I believe that there was a misunderstanding on how the results of this section relate to the previous literature (see response to comments # 5 and # 6). To my knowledge these are new contributions and remain an important part of this work that should be part of the main text.

Notice that x_m and y_m are undefined when they appear, only later I think I understand what their role is.

I now define these variables also in the caption of Figure 4 (now Figure 3 in the revised version).

In the example given, although measurement dependence and signaling are very related (see e.g. <https://arxiv.org/abs/2105.09037>), I would not say that "Yanina can signal to Xavier". The correct sentence appears later, the parties cannot choose their inputs individually (more strikingly: the parties have no free will).

Thank you for pointing this out, I have substantially improved the explanations in the first part of Section IV (now the first part of Section II B). Essentially, there is an inconsistency between having parties who can (partially) control a jamming mechanism and those parties sharing relativistically causal correlations (i.e. correlations that don't allow the parties to send superluminal signals to others). There are then two options: reject that the parties have control over a jamming mechanism (which we can understand as restricting their "free will") or requiring such control but accepting that some parties can send superluminal signals to others. That said, it is indeed not Yanina who can directly be seen to signal but Charlie, I have corrected this in the new version of the manuscript.